# Intraductal Carcinoma of the Prostate: To Grade or Not to Grade

**DOI:** 10.3390/cancers15225319

**Published:** 2023-11-07

**Authors:** Jerasit Surintrspanont, Ming Zhou

**Affiliations:** 1Department of Pathology, King Chulalongkorn Memorial Hospital, The Thai Red Cross Society, Bangkok 10330, Thailand; jerasit.s@chula.ac.th; 2Special Task Force for Activating Research (STAR), Department of Pathology, Chulalongkorn University, Bangkok 10330, Thailand; 3Department of Pathology and Laboratory Medicine, Tufts Medical Center, Boston, MA 02111, USA

**Keywords:** intraductal carcinoma of the prostate, IDC-P, precursor, in situ, GUPS, ISUP, recommendation, reporting practice, basal cell markers, immunohistochemistry

## Abstract

**Simple Summary:**

Whether intraductal carcinoma of the prostate (IDC-P) should be factored into prostate cancer (PCa) Grade Group (GG) has been debated vigorously. While there is currently no study that directly compares the two grading approaches with IDC-P incorporated in or excluded from GG, evidence to date suggests that IDC-P should generally be graded to avoid undergrading and to minimize the overuse of basal cell stains, since >95% of IDC-P represents a high-grade PCa with retrograde spread, which has aggressive biological behavior and is also molecularly akin to high-grade PCa. The only exception to this rule is when IDC-P is present in association with GG1 PCa or in pure form without concomitant PCa, which is a rare scenario and accounts for ~2% of IDC-P. IDC-P in such context could represent a precursor lesion with a prognosis similar to GG1 PCa when diagnosed in radical prostatectomy, and hence should not be included in GG to avoid overgrading.

**Abstract:**

Intraductal carcinoma of the prostate (IDC-P) is a distinct tumor type characterized by an expansile growth of atypical glandular epithelial cells within pre-existing prostate glands and ducts and has significant implications on clinical outcomes and patient management. There is an agreement that isolated IDC-P should not be graded, and IDC-P should be reported with a comment on its clinical significance. However, whether IDC-P should be factored into Grade Group (GG) in the presence of concurrent prostate cancer (PCa) has been debated vigorously. The contradicting opinions were promulgated when the Genitourinary Pathology Society (GUPS) and the International Society of Urological Pathologists (ISUP) published their recommendations for this issue. When IDC-P is present with PCa, the ISUP recommends incorporating it in the GG for the entire case, whereas the GUPS recommends excluding it from the final GG. Consequently, pathologists and clinicians are faced with the conundrum of conflicting recommendations. In this review article, the authors evaluate the magnitude of discrepant GG between the two grading methods, explore the rationales behind the differing views of the two urological societies, present the current reporting practices for IDC-P, and propose a provisional and pragmatic guide to alleviate the dilemma of which recommendation to follow.

## 1. Introduction

Intraductal carcinoma of the prostate (IDC-P) was formally recognized by the World Health Organization (WHO) Blue Book in 2016 as a distinct tumor type [1] and is defined in the 2022 WHO Blue Book as “A neoplastic epithelial proliferation involving pre-existing, generally expanded, duct-acinar structures and characterized by architectural and cytological atypia beyond what is acceptable for high-grade prostatic intraepithelial neoplasia (HGPIN)” [2]. Over the years, numerous studies have demonstrated that IDC-P negatively and profoundly impacts the prognosis of prostate cancer (PCa). With regard to management, IDC-P is a contraindication for active surveillance even as an isolated finding or with Grade Group (GG) 1 PCa [3], and germline genetic testing for mismatch repair and homologous DNA repair should also be considered [2].

There is a general agreement that isolated IDC-P should be reported with a comment on its clinical significance, but not graded. However, whether IDC-P should be graded and factored into GG in the presence of concomitant PCa has been debated vigorously [4,5,6,7,8,9,10,11,12,13,14]. The contradicting opinions were promulgated when the Genitourinary Pathology Society (GUPS) and the International Society of Urological Pathologists (ISUP) independently published their consensuses on this issue [15,16]. While both societies concurred on not grading isolated IDC-P without a concomitant PCa, the GUPS recommended not grading IDC-P in the presence of a concomitant PCa and to perform basal cell marker (BCM) immunohistochemistry (IHC) to differentiate IDC-P from PCa if the result would change the highest GG of the case [15]. The ISUP, on the contrary, recommended incorporating IDC-P in GG if there is a concomitant PCa, which eliminates the need for BCM IHC [16]. The 2022 WHO Blue Book does not endorse either position as both were based primarily on consensus opinions without any conclusive evidence [2]. Consequently, many practicing pathologists and clinicians are faced with the conundrum of these two conflicting recommendations.

In this review article, we will first evaluate the magnitude of discrepant GG between grading and not grading IDC-P, explore the major rationales behind the differing recommendations of the GUPS and the ISUP, present the current state of reporting practices for IDC-P, and propose a provisional and pragmatic guide to alleviate pathologists’ dilemma of which recommendation to follow. The views and opinions expressed in this article are the authors’ own and do not reflect those of the GUPS or the ISUP.

## 2. Magnitude of Discrepancy between Grading and Not Grading IDC-P

The incidence of IDC-P is estimated to be 2.8% of all, including benign and malignant, needle biopsy (NBx) cases [17], or 11–14% of NBx cases positive for PCa [17,18]. In radical prostatectomy (RP) specimens, the reported incidence is higher, ranging from 15.4 to 31.1% [2]. Despite the presence of IDC-P in a significant % of positive NBx and RP specimens, at least three studies have consistently demonstrated that grading or not grading IDC-P had a minimal impact on GG assignment [18,19,20].

### 2.1. Magnitude of Discrepant GG

The study by Chen-Maxwell et al. is often cited to support that grading or not grading IDC-P would change the GG in as many as 23% (28/123) of NBx cases [19]. A cursory glance at this figure can be misleading as the denominator was the number of cases containing GG1–4 PCa with IDC-P, not the total number of cancer cases. This study included 4630 [18] NBx cases, of which 2726 had PCa, and 123 had invasive GG1–4 PCa with IDC-P. Cases with GG5 PCa with IDC-P were not included in the analysis as they already had the highest GG and would not be affected by IDC-P. The GG discrepancy between the two grading methods would be 1% (28/2726) for NBx with PCa and only 0.6% (28/4630) for all NBx.

Rizzo et al. conducted a similar study, which also included only GG1–4 PCa with IDC-P, and obtained an identical result as Chen-Maxwell et al. when the highest GG was used [21]. They also found that the discrepancy was slightly higher for global GG [21]. However, the denominator did not include GG5 PCa or PCa without IDC-P. Because the total number of cases with and without IDC-P was not provided, we were not able to recalculate the GG discrepancy between the two grading methods for NBx with PCa and all NBx. Of note, two other studies by Rijstenberg et al. and Tzelepi et al. reported results very similar to our recalculated results of Chen-Maxwell’s study, with the discordant rate of 1.6% and 0.6% for positive NBx and RP in Rijstenberg’s study [18] and the discordant rate of 1.6% for RP in Tzelepi’s study [20].

Overall, the discordance of GG between these two grading recommendations is merely 1–1.6% for NBx with PCa and 0.6–1.6% for RP. Therefore, GG would be concordant in almost all cases regardless of the grading methods. In discordant cases, grading IDC-P invariably yielded higher GG than not grading IDC-P with a GG difference ranging from 1 to 4. Grading IDC-P resulted in a lower GG in only one case when there was abundant micropapillary IDC-P, which was graded as Gleason pattern (GP) 3 [21]. Hence, whether or not IDC-P should be graded and incorporated in GG is more of an academic discussion. Data from the four studies are tabulated and summarized in Appendix A.

### 2.2. Magnitude of Discrepant Risk Category

From a broader perspective, GG is one of the several clinicopathological factors, including clinical stage, PSA, and the percentage of cancer in each core, used for risk stratification. Rizzo et al. not only looked at the discrepant GG between the two grading methods but also evaluated the difference in National Comprehensive Cancer Network^®^ (NCCN^®^) risk groups. Of the 11 cases with discordant GG, only 36% (4/11) had discrepant NCCN risk categories with all 4 cases falling under higher risk groups by 1 or 3 categories if IDC-P was graded [21]. When this result is extrapolated, the % of cases in the published studies that would have different NCCN risk categories due to the different grading methods of the GUPS and the ISUP is 36% of 1.6%, approximately only 0.6%. Once a patient is placed in a risk group, it is conceivable that there are even more factors, such as life expectancy and competing comorbidities, that also play a role in determining the management options for a patient.

In summary, the two grading recommendations yield little difference in terms of GG assignment and even less so for risk stratification and treatment options.

## 3. Rationales for Grading and Not Grading IDC-P

The decision to grade IDC-P should be based primarily on whether a grading method provides a more accurate prognostic prediction, and secondarily on several practical considerations, including (1) can IDC-P and PCa be reliably distinguished, and (2) how should IDC-P’s prognostic value be conveyed to treating physicians?

### 3.1. Biological and Genetic Basis for Grading and Not Grading IDC-P

There is clinical, biological, and genetic evidence for two distinct pathogenic pathways of IDC-P: the regular or retrograde spread pathway, and the precursor or in situ pathway [2]. The former accounts for most cases and represents a peculiar form of aggressive PCa with retrograde spread into pre-existing ducts and acini, and its grading together with PCa is appropriate. Nevertheless, there are also exceptional situations where IDC-P may represent a precursor or in situ lesion, and it may be prudent to not grade IDC-P. These two types of IDC-P are morphologically indistinguishable, and they are herein referred to as usual-type and precursor-type IDC-P.

#### 3.1.1. Precursor-Type IDC-P

Although exceedingly rare, IDC-P has been observed in association with only GG1 PCa, distant from high-grade PCa, or even without a concurrent PCa (Figure 1) in rare RP specimens; and in these scenarios, IDC-P may represent a precursor lesion in prostate carcinogenesis rather than retrograde spread of high-grade PCa into pre-existing ducts and acini (see discussion below). IDC-P identified in these contexts have been reported in literature by various institutions [18,22,23,24,25,26,27,28], though some of these reports are not as convincing due to partial submission of RP specimens. A summary of the reported cases is provided in Appendix A.

The true incidence of isolated IDC-P in RP is not known but undoubtedly very low with only case reports in the literature, whereas the incidence of IDC-P with GG1 PCa in RP is in the range of 0–0.24% [18,20]. The combined incidence of isolated IDC-P and IDC-P with GG1 PCa is <1% in RP. The incidence of isolated IDC-P in NBx is from 0.06% [29] to 0.26% [17] of all NBx or 1% of positive NBx [17], while the incidence of IDC-P with GG1 PCa in NBx is 0.09% of all NBx [19] and 0.15–0.39% of positive NBx [18,19]. Even if only the specimens with IDC-P are considered, the precursor-type IDC-P still makes up only a tiny fraction of IDC-Ps, with isolated IDC-P accounting for 1.7% of IDC-P in NBx [17,18], and IDC-P with GG1 PCa accounting for 2.3% in NBx and 1.4% in RP [17,18,20] (Table 1).

Precursor-type IDC-P is biologically and genetically different from the usual-type IDC-P. Khani et al. demonstrated that a subset of IDC-P seen in the absence of PCa or concurrent with GG1 PCa shows a unique *MAPK/PI3K* mutational profile distinct from high-grade PCa [26]. In addition, the PTEN and ERG expression patterns are discordant between IDC-P and the accompanying PCa in some cases, suggesting that at least some of these IDC-Ps represent a precursor lesion and are unlikely to result from the retrograde extension of the concomitant PCa [26].

The prognostic significance of the precursor-type IDC-P is also distinct from the usual-type IDC-P. While the latter is associated with poor outcomes, the precursor-type IDC-P, when diagnosed in RP, has a drastically better prognosis with a 5-year biochemical recurrence (BCR)-free survival rate of 93% [24], comparable to GG1 PCa [30]. However, the precursor-type IDC-P has a very different significance when diagnosed in NBx. Although it is associated with unsampled PCa with a GG2 or above in the majority of cases, there is a small but definite chance, approximately 10% for isolated IDC-P and 20% for IDC-P with GG1 PCa, that no invasive PCa or only GG1 PCa will be found in the subsequent RPs [23,25]. Grading of the precursor-type IDC-P in such cases may result in overgrading and inflated GG, and can potentially lead to overtreatment, i.e., unnecessarily subjecting some patients to upfront adjuvant therapy [12]. If isolated IDC-P is not graded per recommendations of both the GUPS and the ISUP because a small subset could represent a precursor lesion, IDC-P with GG1 PCa, which has an even greater probability of being a precursor lesion, also should not be graded for the same reason.

#### 3.1.2. Usual-Type IDC-P

The vast majority of IDC-P is found in association with GG ≥ 2 PCa and is considered to represent a retrograde spread or colonization of benign glands by PCa (Figure 2). In support of this argument, IDC-P shares genetic alterations with high-grade PCa demonstrated by many studies using IHC, fluorescence in situ hybridization, loss of heterozygosity, and other molecular techniques [31]. Examples of these molecular resemblances include loss of PTEN expression, ERG overexpression, and genomic instability [2]. Like aggressive PCa, IDC-P harbors somatic copy-number changes involving *PTEN*, *CDH1*, *BCAR1*, and *MYC*; and mutations in genes such as *SPOP*, *TP53*, and *FOXA1* [2]. It also expresses *SCHLAP1*, a long non-coding RNA associated with a poor prognosis [2]. When tested with genomic risk classifiers, cribriform carcinoma and IDC-P more commonly have higher Oncotype DX Genomic Prostate and Decipher Prostate Cancer Test scores, which predict more adverse outcomes [31]. These molecular similarities between IDC-P and high-grade PCa argue for them to be included in GG when they are present together.

IDC-P is a significant predictor of adverse pathological features of PCa and clinical outcomes independent of other known clinicopathological factors. The presence of IDC-P in NBx strongly correlates with early BCR, cancer-specific survival, survival in patients with distant metastasis at presentation, and post-radiation metastatic failure in intermediate- and high-risk PCa [32,33]. In RPs, IDC-P is strongly associated with adverse pathological features, including a higher grade, larger tumor volume, and greater probability of extraprostatic extension, seminal vesicle invasion, and pelvic lymph node metastasis; as well as adverse clinical outcomes, including early BCR, progression-free survival, and cancer-specific mortality following RP [34,35,36,37,38]. Thus, the WHO currently recommends the reporting of IDC-P, and germline *BRCA2* testing has been recommended by the NCCN and the Philadelphia Prostate Cancer Consensus Conference [39,40]; although, none of the commonly used risk stratification models incorporates IDC-P in the prediction. Its inclusion in GG in the presence of GG ≥ 2 PCa can therefore address, at least partially, the adverse prognostic impact of IDC-P that is currently missing in these risk stratification algorithms.

### 3.2. Improved Prognostication for GG: Grade or Not Grade IDC-P?

The best way to decisively establish which grading method will produce a more accurate prognostic model is to test them in a head-to-head comparison study. Such studies are unfortunately nonexistent at the present time. Nonetheless, published studies have provided some clues regarding which method may provide more accurate prognostic information.

The study by Wang et al. was the only study to our knowledge that directly compared the two grading methods [41], but the result still did not resolve the ongoing debate. The cohort consisted of 558 RP specimens with GG2–5 PCa, of which 38% (213/558) had IDC-P. GG was assigned according to both the GUPS and the ISUP recommendations, and GG2 was used as the baseline. Overall, the hazard ratio for progression-free survival of each individual GG3–5 was significantly different from that of GG2 but not in a fully stepwise trend whether IDC-P was included in GS or not. All GGs in the study, GG2 with tertiary GP5, GG3, GG3 with tertiary GP5, GG4, and GG5, had significantly worse outcomes if graded with the GUPS method; whereas only the last three had significantly worse outcomes if graded with the ISUP method. Regrettably, the investigators did not calculate Harrell’s concordance indices to determine which of the two grading methods has a better discriminative value.

An important argument for IDC-P to be included in GS is that it is very plausible that IDC-P was graded as either GP4 or GP5 in historical and contemporary prognostic studies [30]. Even though these studies did not explicitly state that, this assumption is reasonable. It was noted that the 2000 Armed Force Institute of Pathology Atlas of Tumor Pathology and the 2004 WHO Blue Book contain images of comedonecrosis with morphologically very convincing basal cells [42]. Moreover, two recent studies also showed that foci of comedonecrosis in RP specimens are mostly either IDC-P or an admixture of IDC-P and invasive PCa, and are rarely ever pure invasive PCa [42,43]. Should IDC-P be excluded from grading, such practice change should be supported by data, analogous to the inclusion of a minor high-grade component in the NBx GG, which was based on consensus [44] and supported by studies [45,46]. Therefore, the practice of including IDC-P in the grading should remain unaltered before any compelling evidence becomes available.

Another reason favoring the inclusion of IDC-P in PCa grading is that it may better predict clinical outcomes, and exclusion of IDC-P from PCa grading may lead to undergrading. A study by Kato et al., which only graded PCa by excluding IDC-P but noted its presence, demonstrated that the 10-year BCR-free survivals of GG2–5 PCa with IDC-P are significantly worse than GG2–5 PCa without IDC-P for all GG categories [47], implying that PCa with IDC-P would be undergraded with falsely estimated better prognosis if one conscientiously excludes IDC-P from grading. Similarly, GG3–5 PCa without IDC-P in Kato’s study [47] had better 10-year BCR-free survivals than those in Epstein’s GG validation study [30], 62% vs. 49%, 49% vs. 37%, and 43% vs. 16%, respectively; the latter study was assumed to include IDC-P in the GG. This comparison suggests that inclusion of IDC-P in the grading would result in a more accurate prognosis prediction. A recent study by van Leenders et al. also showed “cribriform grade (cGrade),” which incorporates invasive cribriform carcinoma and/or IDC-P in the Gleason grading, provides a better discriminative power for clinical outcomes than the GG, particularly impacting decision making in men with current GG2 PCa [9].

The counterargument is that the number of cases that GG would be affected by grading or not grading IDC-P are too few to make any statistical impact on the overall prognostic correlation [12], i.e., had IDC-P been excluded from grading, GG would still correlate with the prognosis, indirectly implying that excluding IDC-P from GS should also be a valid method. In support of this argument, studies have shown that grading or not grading IDC-P has only a marginal effect on GG assignment [18,19,20]; and GG with IDC-P included or not, both correlated with BCR [47], albeit not entirely in an incremental fashion.

Because there are currently no studies to prove that excluding IDC-P from GG is superior, the authors believe that the grading method that incorporates IDC-P in GG, presumably used in historical and contemporary datasets, should continue to be used. By doing so, one can incorporate IDC-P, a significant adverse pathological parameter, in the risk stratification tools to avoid undergrading and undertreating biologically aggressive PCa, ensure correct prognostication, and obviate the excessive use of BCM IHC (see discussion below).

### 3.3. Practical Issues Related to Grading and Not Grading IDC-P

A myriad of other issues has also been brought up in both sides in previous debates to bolster the argument for including or not including IDC-P in the GS and is compiled and addressed in this subsection. The supporting evidence of these arguments, though, is less robust and on no account does their combined weight surpass that of the rationales detailed in Section 3.1 and Section 3.2.

#### 3.3.1. BCM IHC

There is a concern for excessive use of BCM IHC if IDC-P is to be distinguished from invasive PCa and excluded from GG. The distinction between IDC-P from invasive PCa can be difficult if not impossible on routine H&E stain in some cases. 65% of ISUP survey respondents reported that IDC-P cannot be reliably diagnosed without BCM IHC [16]. Chen-Maxwell et al. used clearly visible basal cells as the only criterion to differentiate IDC-P from PCa and performed BCM IHC in >80% of the cases containing IDC-P with GG1–4 PCa [19]. In another study, Rijstenberg et al. found that BCM IHC was required to discern the two in roughly 50% of positive NBx and RP specimens [18].

Although the GUPS endorses excluding IDC-P from GS, it also clearly stated that performing BCM IHC to differentiate IDC-P from invasive PCa is recommended only in cases that the result would change the highest GG for the case [15]. For example, performing BCM IHC is justifiable when one is deciding between GG2 and GG3 PCa [12]. It is deemed an uncommon occurrence and affects as few as 0.7% of PCa cases in Epstein’s anecdotal experience [12]; however, this number could be much higher for non-academic pathologists. Seemingly simple, this recommendation is criticized by some as being too complicated because it requires determining first whether atypical glands with cribriform, comedonecrosis, or solid growth pattern could represent IDC-P, followed by gauging if excluding IDC-P would change the GG [12].

There are additional arguments against using BCM IHC routinely to distinguish IDC-P from invasive PCa. There are no studies to show that IDC-P by morphology and by IHC differ in outcomes [12]. Furthermore, there is significant variation in the interpretation of BCM IHC and that IDC-P cannot be entirely excluded with a negative BCM IHC. For instance, a cluster of comedonecrosis with basal cells admixed with rare glands without basal cells is regarded as mixed IDC-P and invasive PCa by some [42] and entirely as IDC-P by others [43]. The latter interpretation takes into consideration that a small IDC-P gland may be entirely negative for BCM IHC as the result of outpouching or tangential sectioning of an adjacent large IDC-P, analogous to HGPIN with adjacent small atypical glands. In addition, expert genitourinary pathologists may include IDC-P in GS but may exclude it from GS if BCM IHC is available for review [3]. Finally, variable usage of BCM IHC and applicability in limited resource settings are among the potential problems if BCM IHC is to be used per GUPS recommendation [12].

#### 3.3.2. Inclusion of IDC-P to Improve PCa Risk Stratification Models

The NCCN Clinical Practice Guidelines in Oncology (NCCN Guidelines^®^), the European Association of Urology (EAU) Guidelines as well as most other nomograms currently incorporate pathological findings such as GG and tumor volume but not IDC-P for risk stratification. Grading IDC-P with invasive PCa would include IDC-P, a significant adverse morphological feature of PCa, in GG; therefore, it is a more reliable way of conveying IDC-P’s important prognostic value and ensuring that it is incorporated into the risk stratification tools and treatment plan. This is analogous to the inclusion of tertiary GP5 in NBx as the secondary GP in GG to avoid omission by clinicians should it be only documented in a comment, a recommendation endorsed at the 2005 ISUP consensus conference [12,44]. Similarly, most cancer registries do not document the presence of IDC-P, and grading IDC-P can help capture this crucial data point [12].

#### 3.3.3. Other Miscellaneous Issues

Since most pathologists consider IDC-P as part of the tumor extent, including IDC-P in GS makes it simpler and more consistent to estimate the % of GP4 and tumor extent [12].

Even the opponents of grading IDC-P agreed that it is reasonable to grade IDC-P in the setting of overt PCa [12], leading to an inevitable conclusion that IDC-P could be included in the GG in those cases.

A concern was raised that grading IDC-P may increase the likelihood that IDC-P will not be reported, which may affect treatment [12]. However, proponents of grading IDC-P also recommend the reporting of IDC-P in addition to factoring it in the GG.

The opponents of grading IDC-P also argued that by not grading IDC-P, future studies can be conducted to compare the two grading methods [12]. Since reporting IDC-P is recommended in pathology reports regardless of the grading method, it is possible to retrieve cases with IDC-P included in the GG for comparative studies afterwards.

## 4. Current State of Reporting IDC-P and Use of BCM IHC in the Diagnosis of IDC-P

Since the official recognition of IDC-P as a distinct entity by the WHO Blue Book in 2016, several surveys have been conducted to evaluate the reporting practices and usage of BCM IHC related to the diagnosis of IDC-P [3,15,16,48,49]. The survey by Williamson et al. [49] was conducted after the publications of the GUPS and the ISUP consensus papers [15,16] and should reflect the most up-to-date practice trend in the era of the recommendations from both urological societies. The survey results relevant to the topic of this article are summarized in Table 2 [49].

Most pathologists do not grade IDC-P in both NBx and RP specimens, but are more likely not to grade isolated IDC-P than IDC-P with GG1 PCa and IDC-P with GG ≥ 2 PCa [49]. 90% of pathologists would not grade isolated IDC-P in both NBx and RP specimens. In the presence of invasive PCa, 72% would not grade IDC-P with GG1 PCa in both NBx and RP; and 59% and 57% would not grade IDC-P with GG ≥ 2 PCa in NBx and RP, respectively. Similarly, an earlier survey of 42 international urological pathology specialists [3] conducted before the publication of the recommendations by GUPS and ISUP found that morphologically recognizable IDC-P, when present with GG1 PCa, was not included in the GS in NBx (78%) or RP (71%) specimens. The majority did not grade IDC-P with comedonecrosis in NBx (62%) or RP (69%) specimens. However, most of the respondents (60%) would include readily recognizable IDC-P in the assessment of linear extent of PCa in NBx.

The decision to use BCM IHC is situational [49], i.e., most pathologists would utilize BCM IHC when the precursor-type IDC-P is a diagnostic consideration but not for the usual-type IDC-P. 92% of respondents would use BCMs to resolve the differential diagnosis of IDC-P versus invasive cribriform PCa when no definite invasive PCa is identified in NBx. 84% and 72% of respondents would do BCM IHC when GG1 PCa is seen together with IDC-P in NBx and RP, respectively. In contrast, only 15% and 21% of respondents would use BCM IHC if GG ≥ 2 PCa is found in association with IDC-P in NBx and RP, respectively. Comparably, an earlier survey [3] reported that a significant majority (78%) would use IHC to confirm or exclude IDC-P if NBx showed no PCa.

Additionally, approximately 60% of pathologists would use IHC to confirm IDC-P with invasive PCa in NBx if it would change the overall GS [3]. Nearly half (48%) would use IHC to confirm IDC-P for accurate GP4 quantitation [3]. Although IHC for PTEN and/or ERG has been proposed to aid the diagnosis of IDC-P [50], most (75%) did not use them to distinguish IDC-P from HGPIN in NBx [3].

The majority opinion from these surveys favors not to grade IDC-P when it is an isolated finding or concomitant with GG1 PCa in both NBx and RP specimens, and BCM IHC be used in these settings to confirm the IDC-P diagnosis.

## 5. The Authors’ Recommendations

To recapitulate, IDC-P is a morphologically defined tumor type with two distinct pathogenic pathways: precursor-type and usual-type. Because these two types are biologically, genetically, and prognostically different, the authors believe that they warrant different grading approaches (Table 3).

Precursor-type IDC-P is rare, accounting for only a few % of all IDC-P, but can occur in a few settings: pure form, with coexisting GG1 PCa, and distant from high-grade PCa. When an entirely submitted RP specimen contains only the precursor-type IDC-P, the prognosis is analogous to GG1 PCa. When diagnosed in NBx, the precursor-type IDC-P indicates unsampled high-grade, high-volume PCa in most cases. However, this is not always the case as 10–20% of NBx with this finding have been shown to have no invasive PCa or only GG1 PCa at RP. Therefore, one should not grade IDC-P in this setting to prevent overgrading, which may lead to overtreatment. In such cases, it is advised to perform BCM IHC to confirm the IDC-P diagnosis and exclude GP4/5 invasive PCa.

On the contrary, usual-type IDC-P is vastly more common than the precursor-type, constituting almost all IDC-Ps in both NBx and RP. This form of IDC-P invariably coexists with GP4 and/or GP5 in GG2–5 PCa. Although a slight majority of pathologists currently do not grade IDC-P in this setting [49], we recommend grading it together with PCa and also including it in the tumor volume measurement, since both usual-type IDC-P and high-grade PCa are comparable as to their biological aggressiveness, molecular alterations, and adverse prognosis. Performing BCM IHC in these cases is optional but not recommended as these stains make only a minor difference with respect to GG assignment.

Regardless of whether IDC-P is graded, its presence and clinical significance should be clearly documented in the pathology reports.

## 6. Summary

The divergent recommendations from the ISUP and the GUPS regarding grading IDC-P have not yet been reconciled as there is currently no study that is specifically designed to directly compare the two grading methods. Such studies should be conducted in the future to clearly delineate the value of including IDC-P in the grading of PCa. Additionally, it would be interesting to investigate the clinical and pathological features that may have a more significant impact than IDC-P grading for treatment decisions and patient outcomes. In a recent survey, most pathologists do not grade IDC-P in NBx and RP specimens and would perform BCM IHC when isolated IDC-P or IDC-P with GG1 PCa is suspected but not when IDC-P is present with GG ≥ 2 PCa.

Based on the available evidence to date, the authors conclude that, as a general rule, it is more scientifically sound and pragmatic to incorporate IDC-P in GG to avoid undergrading and to minimize the use of costly BCM IHC since >95% of IDC-P represents a high-grade PCa with retrograde spread, which has aggressive biological behavior and is also molecularly akin to high-grade PCa. The only exception to this rule is when IDC-P is present in association with GG1 PCa or in pure form without concomitant PCa, which is a rare scenario and accounts for ~2% of IDC-P. IDC-P in such context could represent a precursor lesion with a prognosis similar to GG1 PCa when diagnosed in RP, and 10–20% is associated with finding only IDC-P or GG1 PCa in RP when diagnosed in NBx, and hence should not be included in GG to prevent overgrading. BCM IHC should be performed in latter cases to ascertain the diagnosis and to exclude GP4/5 PCa. Nevertheless, the grading of IDC-P has no impact on patient management in most cases as it has only an almost negligible impact on GG assignment regardless of the grading method used and even less so for risk stratification and treatment decision due to the influence of other clinical and pathological variables.

## Figures and Tables

**Figure 1 cancers-15-05319-f001:**
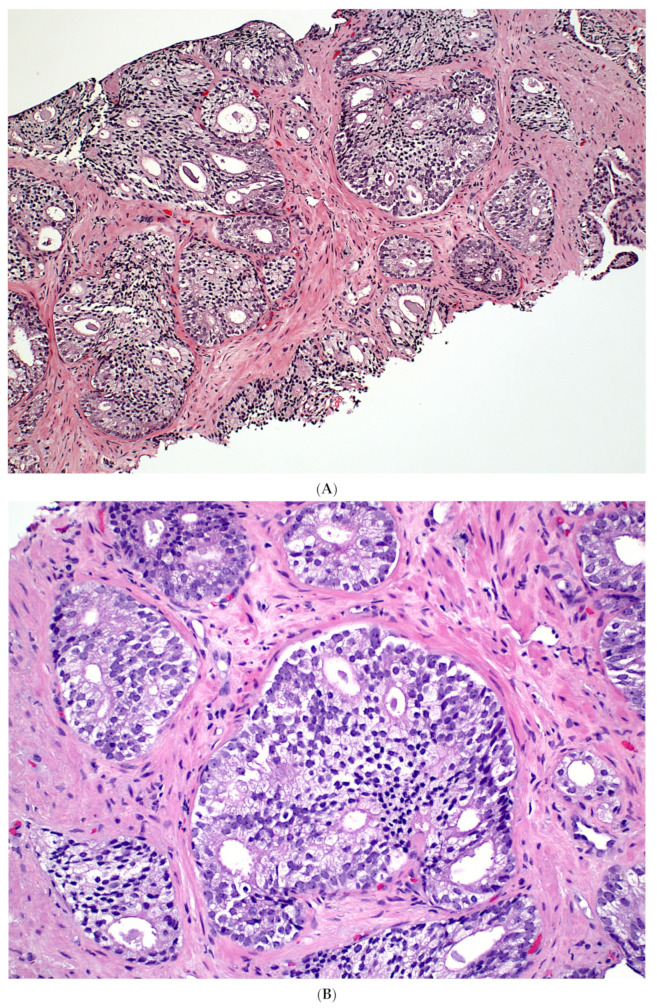
Isolated IDC-P without concomitant prostate cancer in a prostate biopsy (**A**,**B**). The IDC-P glands have residual basal cells positive for basal cell markers (brown stains) and are also positive for AMACR (red stain) (**C**). Isolated IDC-P is associated with unsampled GG ≥ 2 PCa in the majority of cases, although no invasive or only GG1 PCa is found in subsequent RPs in approximately 10% of cases.

**Figure 2 cancers-15-05319-f002:**
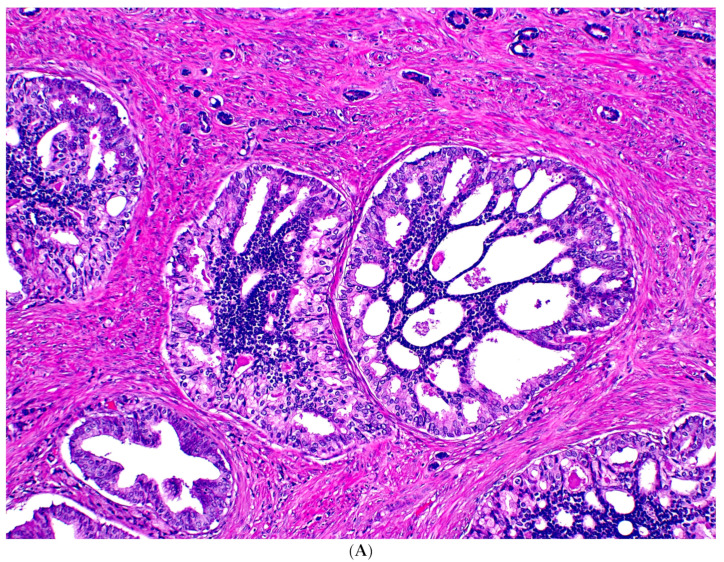
Usual-type IDC-P with GG5 prostate cancer. IDC-P glands show cribriform morphology and are surrounded by invasive cancer glands (**A**). IDC-P glands have residual basal cells highlighted by basal cell marker P63 (brown stain), while the cancer glands lack basal cells (**B**).

**Table 1 cancers-15-05319-t001:** Incidence of precursor-type IDC-P.

Authors, Year	Type of Specimen	# of Cases				
		All	With PCa	With IDC-P	Isolated IDC-P	IDC-P with GG1 PCa
Guo, 2006 [29]	NBx	~45,000	N/A	N/A	27	N/A
Watts, 2013 [17]	NBx	1176	312	33	3	0
Chen-Maxwell, 2020 [19]	NBx	4630	2726	123 (not including IDC-P with GG5 PCa)	0	4
Rijstenberg, 2020 [18]	NBx	N/A	1031	139	0	4
	RP (entirely submitted)	N/A	835	213	0	4
Rizzo, 2021 [21]	NBx	N/A	N/A	48 (not including IDC-P with GG5 PCa)	0	3
Tzelepi, Cancers, 2021 [20]	RP (partially submitted)	N/A	129	81	0	0
Incidence of IDC-P	NBx (pooled from [17,18])	-	1343	172	3 (3/172 = 1.7%)	4 (4/172 = 2.3%)
	RP (pooled from [18,20])	-	964	294	0	4 (4/294 = 1.4%)

GG, Grade Group; IDC-P, intraductal carcinoma of the prostate; N/A, not available; NBx, needle biopsy; PCa, prostate cancer; RP, radical prostatectomy.

**Table 2 cancers-15-05319-t002:** The most recent survey on grading of IDC-P and usage of BCM IHC [49].

		Isolated IDC-P	IDC-P with GG1 PCa	IDC-P with GG ≥ 2 PCa
Grading of IDC-P	NBx	90% do not grade IDC-P	72% do not grade IDC-P	59% do not grade IDC-P
	RP	90% do not grade IDC-P	72% do not grade IDC-P	57% do not grade IDC-P
BCM IHC	NBx	92% use BCM IHC	84% use BCM IHC	15% use BCM IHC
	RP	N/A	72% use BCM IHC	21% use BCM IHC

BCM, basal cell marker; GG, Grade Group; IDC-P, intraductal carcinoma of the prostate; IHC, immunohistochemistry; N/A, not available; NBx, needle biopsy; PCa, prostate cancer; RP, radical prostatectomy.

**Table 3 cancers-15-05319-t003:** The authors’ recommendations for grading IDC-P.

		Precursor-Type IDC-P		Usual-Type IDC-P
		Isolated IDC-P	IDC-P with GG1 PCa	IDC-P with GG2–5 PCa
Incidence	NBx	1.7% of NBx with IDC-P	2.3% of NBx with IDC-P	>95% of NBx with IDC-P
	RP	True incidence not known in RP but probably < IDC-P with GG1 PCa	1.4% of RP with IDC-P	>95% of RP with IDC-P
Biological pathway		~10% of NBx with isolated IDC-P could represent a precursor lesion	~20% of NBx with IDC-P with GG1 PCa could represent a precursor lesion	Retrograde spread of high-grade PCa
Prognosis	NBx	10–20% have no invasive PCa or only GG1 PCa at RP		Equivalent to GG4–5 PCa
	RP	Equivalent to GG1 PCa		
Grading Rule for IDC-P		Do not grade		Include in GG
BCM IHC		Yes, to confirm the diagnosis by excluding GP4 and GP5 PCa		Optional, but not recommended

BCM, basal cell marker; GG, Grade Group; GP, Gleason pattern; IDC-P, intraductal carcinoma of the prostate; IHC, immunohistochemistry; NBx, needle biopsy; PCa, prostate cancer; RP, radical prostatectomy.

## Data Availability

All references cited in this review article are available in WHO Classification of Tumours online, PubMed, and ResearchGate.

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
