# Peer review of "Intraductal Carcinoma of the Prostate: To Grade or Not to Grade"

_cancers, 2023, doi:10.3390/cancers15225319_

Round 1

Reviewer 1 Report

Comments and Suggestions for Authors

Overall, this manuscript is well-written and comprehensive regarding the integration of IDC-P into the group grading system. This reviewer comments on several minor issues.

The authors present Figure 1 as a representative photo of a precursor-type. However, it is difficult to confirm this classification due to the provided needle biopsy case. They need to state that they have confirmed this case was a precursor-type proved by radical prostatectomy.

The authors have presented Figure 2 as a representative photo of a regular-type. However, upon closer inspection, it seems that Figures 2A and 2B demonstrate different areas. Unfortunately, Figure 2A appears to have similar morphological features as Figure 1A, which is a representative precursor-type. Therefore, the authors need to replace Figure 2A to ensure that readers can understand the information easily.

"Conventional-type" or "Usual-type" is more fitting than "Regular-type" as the counterpart of "Precursor-type."

Author Response

We’d like to thank you for your constructive critique. The following is the point-to-point response to their comments.

“The authors present Figure 1 as a representative photo of a precursor-type. However, it is difficult to confirm this classification due to the provided needle biopsy case. They need to state that they have confirmed this case was a precursor-type proved by radical prostatectomy.”

Figures 1A-C are meant to illustrate the morphological features of “precursor-type IDC-P”, which is indistinguishable from usual-type IDC-P.

We modified the figure legend:

Isolated IDC-P without concomitant prostate cancer in a prostate biopsy (A and B). The IDC-P glands have residual basal cells positive for basal cell markers (brown stains) and are also positive for AMACR (red stain) (C). Isolated IDC-P is associated with unsampled PCa with a GG>2 in the majority of cases. However, no invasive PCa or only GG1 PCa is found in the sub-sequent RPs in approximately 10% of cases.

“The authors have presented Figure 2 as a representative photo of a regular-type. However, upon closer inspection, it seems that Figures 2A and 2B demonstrate different areas. Unfortunately, Figure 2A appears to have similar morphological features as Figure 1A, which is a representative precursor-type. Therefore, the authors need to replace Figure 2A to ensure that readers can understand the information easily.”

We apologize for providing the wrong images. We will submit new Figure 2A and B with modified figure legends.

"Conventional-type" or "Usual-type" is more fitting than "Regular-type" as the counterpart of "Precursor-type."

We agree with the review and have changed the “regular-type” to “usual-type”.

Reviewer 2 Report

Comments and Suggestions for Authors

The study of ''Intraductal Carcinoma of the Prostate: to Grade or Not to Grade'' is truly novel and rich in information about the IDC-P and how is the current situation with authors' suggestions. I highly recommend the paper for publication. Only I expected to find in the revised form Table to summarize all the biological markers (molecular, genetics, ....) that have implication in pathogenesis and can be used for for grading and not grading IDC-P

Comments on the Quality of English Language

 Moderate editing of English language is required

Author Response

We’d like to thank you for your constructive critique. The following is the point-to-point response to your comments.

“The study of ''Intraductal Carcinoma of the Prostate: to Grade or Not to Grade'' is truly novel and rich in information about the IDC-P and how is the current situation with authors' suggestions. I highly recommend the paper for publication. Only I expected to find in the revised form Table to summarize all the biological markers (molecular, genetics, ....) that have implication in pathogenesis and can be used for for grading and not grading IDC-P”

The molecular markers of IDC-P are covered by another article in this special issue, "Pathogenesis and molecular underpinnings."

Reviewer 3 Report

Comments and Suggestions for Authors

This article discusses the debate over grading intraductal carcinoma of the prostate (IDC-P) and its impact on prostate cancer prognosis and management.

A major revision is necessary to improve clarity, organization, and readability in the article. Key areas include reorganizing content, providing informative introductions, clarifying abbreviations, and addressing grammar and citation issues.

Here are some points for improvement in the provided article:

  • Clarity and Flow:
    • The article should have a clearer and more logical flow. Consider reorganizing the content to make it easier for readers to follow.
  • Introductory Section:
    • The introduction can be more informative. It should provide a brief overview of what IDC-P is and its significance in the context of prostate cancer before diving into the grading debate.
  • Use of Abbreviations:
    • You should introduce abbreviations like IDC-P, WHO, GG, NBx, and RP with their full forms before using them consistently. This will help readers who may not be familiar with these terms.
  • Subheadings:
    • Consider using subheadings to break down the article into different sections. This will make it more organized and easier to navigate.
  • Grammar and Language:
    • There are some grammatical issues, like incomplete sentences and awkward phrasing. Review the text for clarity and coherence.
  • Data Presentation:
    • The article discusses several studies but could benefit from more data visualization, such as tables and figures, to help readers better understand the magnitude of discrepancies and other statistical information.
  • Discussion of Recommendations:
    • When discussing the differing recommendations of GUPS and ISUP, it would be helpful to summarize their key points and reasons for the recommendations in a more structured manner.
    • Please include and discuss the following: PMID: 36294423; PMID: 36363581 
  • Conclusive Recommendations:
    • The article should conclude with clear recommendations or potential directions for future research. Readers should leave with a clear understanding of the authors' stance on the issue.
  • Use of Visuals:
    • Consider including visual aids like diagrams, charts, or images to illustrate concepts, especially when discussing different types of IDC-P.
  • Clarity on Position:
    • Make it clear whether the authors are advocating for a specific position on grading IDC-P or if they are presenting a balanced view of the debate.

Remember to adapt these suggestions to the specific requirements and guidelines of the journal or platform where you intend to publish the article

Author Response

We’d like to thank you for your constructive critique. The following is the point-to-point response to your comments.

“A major revision is necessary to improve clarity, organization, and readability in the article. Key areas include reorganizing content, providing informative introductions, clarifying abbreviations, and addressing grammar and citation issues.

Here are some points for improvement in the provided article:

Clarity and Flow:

The article should have a clearer and more logical flow. Consider reorganizing the content to make it easier for readers to follow.”

This manuscript is structured to facilitate the understanding of the pros and cons of grading IDC-P. We first discussed the magnitude of the discrepancy between grading and not grading IDC-P to set the stage for the discussion. We then laid out the rationales for grading and not grading IDC-P, including the biological and genetic basis, improved prognostication for grade group, and other practical issues related to grading and not grading IDC-P. We also provided an update on the reporting IDC-P and the use of BCM IHC in the diagnosis of IDC-P. Finally, we made recommendations for reporting IDC-Ps based on different clinical scenarios.

“Introductory Section:

The introduction can be more informative. It should provide a brief overview of what IDC-P is and its significance in the context of prostate cancer before diving into the grading debate.”

Since this manuscript is a chapter in a special issue on IDC-P, other chapters will provide an overview of IDC-P and its clinical significance. Therefore we kept the "Introduction" concise.

“Use of Abbreviations:

You should introduce abbreviations like IDC-P, WHO, GG, NBx, and RP with their full forms before using them consistently. This will help readers who may not be familiar with these terms.”

We double-checked and made sure the manuscript complies with the rule of “use of abbreviations”.

“Subheadings:

Consider using subheadings to break down the article into different sections. This will make it more organized and easier to navigate.”

We used subheadings and subsections throughout the manuscript to make it easier to read and navigate.

“Grammar and Language:

There are some grammatical issues, like incomplete sentences and awkward phrasing. Review the text for clarity and coherence.”

We thoroughly checked the manuscript for spelling and grammatical accuracy.

“Data Presentation:

The article discusses several studies but could benefit from more data visualization, such as tables and figures, to help readers better understand the magnitude of discrepancies and other statistical information.”

When discussing the “magnitude of discrepancy between grading and not grading IDC-P”, we collated all the relevant data in the supplemental Table 1 and discussed the most important data in the text with clear explanations for the magnitude of discrepancies.

“Discussion of Recommendations:

When discussing the differing recommendations of GUPS and ISUP, it would be helpful to summarize their key points and reasons for the recommendations in a more structured manner.

Please include and discuss the following: PMID: 36294423; PMID: 36363581 “

We succinctly summarized the key points of ISUP and GUPS recommendations regarding reporting IDC-P in the “Introduction” (page 2).

We are puzzled by the reviewer’s suggestion to include two references that do not seem to be relevant to the topic of this manuscript.

“Conclusive Recommendations:

The article should conclude with clear recommendations or potential directions for future research. Readers should leave with a clear understanding of the authors' stance on the issue.”

Table 3 clearly states authors’ recommendations for grading IDC-P and the use of basal cell marker IHC in the work-up of IDC-P.

“Use of Visuals:

Consider including visual aids like diagrams, charts, or images to illustrate concepts, especially when discussing different types of IDC-P.”

We used 3 tables and 2 supplemental tables to discuss the incidence and clinical significance of precursor-type and usual-type IDC-P.

“Clarity on Position:

Make it clear whether the authors are advocating for a specific position on grading IDC-P or if they are presenting a balanced view of the debate.”

The manuscript provided a balanced view of the rationales behind “grading” vs “not grading” IDC-P. The authors also provide a recommendation for grading IDC-P as a pragmatic guide to alleviate the dilemma of which recommendation to follow.

Reviewer 4 Report

Comments and Suggestions for Authors

The article is well-structured and presents the issue of grading IDC-P in a clear and organized manner, drawing upon the available literature. While the article mentions the absence of a study directly comparing the ISUP and GUPS grading methods for IDC-P, it would be valuable if the authors could discuss the potential benefits of conducting such a study in the future. Additionally, it would be interesting to know more about the clinical and pathological variables that may have a more significant impact than IDC-P grading in terms of treatment decisions and patient outcomes. In general, I find this article to be a valuable resource for understanding the potential future developments in IDC-P grading and their potential impact on clinical practice.

Comments on the Quality of English Language

The English in the article is generally acceptable, with only minor editing needed.

Author Response

We’d like to thank you for your constructive critique. The following is the point-to-point response to your comments.

“Comments and Suggestions for Authors

The article is well-structured and presents the issue of grading IDC-P in a clear and organized manner, drawing upon the available literature. While the article mentions the absence of a study directly comparing the ISUP and GUPS grading methods for IDC-P, it would be valuable if the authors could discuss the potential benefits of conducting such a study in the future. Additionally, it would be interesting to know more about the clinical and pathological variables that may have a more significant impact than IDC-P grading in terms of treatment decisions and patient outcomes. In general, I find this article to be a valuable resource for understanding the potential future developments in IDC-P grading and their potential impact on clinical practice.”

We added the following in the “Summary”:

 ...Such studies should be conducted in the future to clearly delineate the value of including IDC-P in the grading of PCa. Additionally, it would be interesting to investigate the clinical and pathological features that may have a more significant impact than IDC-P grading for treatment decisions and patient outcomes.

Round 2

Reviewer 3 Report

Comments and Suggestions for Authors

the paper has been improved